# Impact of the COVID-19 Pandemic on Solid Organ Transplant and Rejection Episodes in Brazil’s Unified Healthcare System

**DOI:** 10.3390/jcm11216581

**Published:** 2022-11-06

**Authors:** Luis Gustavo Modelli de Andrade, Abner Macola Pacheco Barbosa, Naila Camila da Rocha, Marilia Mastrocolla de Almeida Cardoso, Juliana Tereza Coneglian de Almeida, Juliana Machado-Rugolo, Lucas Frederico Arantes, Daniela Ferreira Salomão Pontes, Gustavo Fernandes Ferreira

**Affiliations:** 1Department of Internal Medicine—UNESP, Univ Estadual Paulista, Av. Prof. Montenegro—Distrito de, Rubião Jr., s/n, Botucatu 18618-687, SP, Brazil; 2Health Technology Assessment Center Hospital das Clínicas—HCFMB, Botucatu 18618-970, SP, Brazil; 3Transplant Unit—Santa Casa Juiz de Fora, Av. Barão do Rio Branco, 3353-Passos, Juiz de Fora 36021-630, MG, Brazil

**Keywords:** COVID-19, organ transplant, graft rejection, Brazil

## Abstract

Background: Brazil has the world’s largest public organ transplant program, which was severely affected by the COVID-19 pandemic. The primary aim of the study was to evaluate differences in solid organ transplants and rejection episodes during the COVID-19 pandemic compared to the five years before the pandemic in the country. Methods: A seven-year database was built by downloading data from the DATASUS server. The pandemic period was defined as March 2020 to December 2021. The pre-pandemic period was from January 2015 to March 2020. Results: During the pandemic, the number of solid organ transplants decreased by 19.3% in 2020 and 22.6% in 2021 compared to 2019. We found a decrease for each evaluated organ, which was more pronounced for lung, pancreas, and kidney transplants. The seasonal plot of rejection data indicated a high rejection rate between 2018 and 2021. There was also an 18% (IRR 1.18 (95% CI 1.01 to 1.37), *p* = 0.04) increase in the rejection rate during the COVID-19 pandemic. Conclusions: The total number of organ transplants performed in 2021 represents a setback of six years. Transplant procedures were concentrated in the Southeast region of the country, and a higher proportion of rejections occurred during the pandemic. Together, these findings could have an impact on transplant procedures and outcomes in Brazil.

## 1. Introduction

Brazil has an extensive public healthcare system known as the Unified Healthcare System (SUS), and public health expenditures in the country represent approximately 9% of the gross domestic product (GDP). The national organ transplant system, which is the largest in the world, is responsible for conducting 95% of all solid organ transplants as well as purchasing immunosuppressants [1].

There are several difficulties in carrying out a comprehensive health information analysis in Brazil, especially considering the size of the country. However, the Brazilian SUS has an excellent information database called DATASUS [2]. The information systems maintained by DATASUS include several aspects of population health, including epidemiological, such as the Mortality Information System (SIM) and the Live Birth Information System (SINASC), which recover data from registry offices, as well as administrative, such as the Hospital Admissions System (SIH) and the Outpatient Information System (SIA), which retrieve data from public healthcare providers [3]. These databases are open source and are made publicly available by the Brazilian government.

The COVID-19 pandemic severely affected transplants worldwide, reducing the number of procedures, particularly in the early phase of the pandemic [4]. The mortality of transplant patients was higher than the general population, reaching 20–25% in the initial stages of the pandemic [5]. Despite this, there was a reduction in mortality rates during the COVID-19 pandemic related to improvements in patient management and vaccination [6].

Changes in the immunosuppression regimen of transplant patients are frequent due to higher rates of hospitalization, need for supplemental oxygen, or admission to the intensive care unit [6]. During the pandemic, medical appointments were postponed, and others were carried out by telemedicine [7]. We hypothesized that changes in immunosuppression combined with the difficulty of attending medical appointments may have increased solid organ rejection rates. Thus, the primary aim of this study was to evaluate the changes in solid organ transplants and rejection episodes during the COVID-19 compared to the five years before the pandemic in Brazil.

## 2. Methods

### 2.1. Population

We used data provided by DATASUS [2], a health data information system maintained by the Brazilian Ministry of Health that is anonymized and publicly available. DATASUS retrieves information about the Unified Healthcare System in Brazil (SUS) and is linked to reimbursement. All public healthcare providers in Brazil must provide monthly electronic reports. Data on procedures are available through the Hospital Information System (SIH) and can be downloaded from the open access repository (https://datasus.saude.gov.br/ (accessed on 2 October 2022)). The data contain the age, sex, race, location, death during hospitalization, length of stay, reimbursement value, International Classification of Diseases (ICD-10) code, and procedure performed (http://sigtap.datasus.gov.br/ (accessed on 2 October 2022)).

### 2.2. Data Retrieved

A seven-year database was built by downloading data from the DATASUS server. The pandemic period was defined as March 2020 to December 2021. The pre-pandemic period was defined as January 2015 to March 2020.

Information for all transplant procedures represented by the following codes were recovered: deceased donor kidney transplant (SUS code: 05.05.02.009-2); living donor kidney transplant (SUS code: 05.05.02.010-6); kidney-pancreas transplant (SUS code: 05.05.02.011-4); pancreas transplant (SUS code: 05.05.02.007-6); bilateral lung transplant (SUS code: 05.05.02.012-2); unilateral lung transplant (SUS code: 05.05.02.008-4); liver transplant (SUS code: 05.05.02.005-0); and heart transplant (SUS code: 05.05. 02004-1). These represent all solid organ transplants performed in Brazil by the public healthcare system.

From the database, rejection after transplant episodes were retrieved, represented by the following ICD-10 codes: kidney transplant rejection (T86.1); heart transplant rejection (T86.2); heart/lung transplant rejection (T86.3); liver transplant rejection (T86.4); and other organ transplant rejection (T86.8). Data on pancreas, kidney–pancreas, and lung rejection was retrieved from the other transplant rejection code combined with secondary ICD-10 codes. The number of rejection episodes was divided by the total number of transplants performed during the same period.

### 2.3. Statistical Analysis

Individual, anonymized data were obtained through the specific microdata SUS package [3] in the R software environment from the information downloaded from the DATASUS server.

Descriptive epidemiological data on transplant numbers and rejection episodes were performed using measures of central tendency (median and percentiles) and frequency. The rejection episodes were presented as a total number and corrected for the number of transplants performed. To compare baseline characteristics between pre-pandemic and COVID-19 periods, we used a Wilcoxon rank sum test for continuous variables and Pearson’s chi-squared test for categorial variables. To test the hypothesis of a higher rejection frequency during the COVID-19 pandemic, we fitted a quasi-Poisson regression using the year and considered the presence of the COVID-19 pandemic as a predictive factor.

The data used for the analysis and the database will be made publicly available in an open repository. We used R version 4.1.2 for all analyses.

## 3. Results

Over the pre-pandemic period, there was a progressive increase in the total number of solid organ transplants performed by SUS in Brazil, reaching a total of 7847 procedures in 2019. During the COVID-19 pandemic, we found a reduction of 19.3% in 2020 and 22.6% in 2021 compared to 2019 (Table 1).

We found a decrease for each evaluated organ (kidney, liver, lung, heart, and pancreas), but this was more pronounced for lung, pancreas, and kidney transplants (Figure 1A,B).

Overall, we evaluated 48,729 solid organ transplants, divided into 38,429 transplants in the pre-pandemic period and 10,300 during the COVID-19 pandemic. The overall median age was 49 (36–58) years, with patients being predominantly male (62%) and white (57%). Kidney transplants were the most common procedure (69%), followed by liver (24%). The death rate during hospitalization was 4.4%. There were no significant differences between pre-pandemic and COVID-19 patient characteristics (Table 2).

The majority of solid organ transplants in Brazil were performed in the Southeast (states of São Paulo, Minas Gerais, Rio de Janeiro), South (Rio Grande do Sul, Parana), and Northeast regions (Ceara, Pernambuco). After the pandemic, transplant activities were concentrated in the Southeast and one state in the South (Parana), with a reduction in the North and Northeast (Figure 2).

The least affected region was the Midwest, which represents a small proportion of transplant procedures in Brazil (Table 3). The Southeast was the second least affected (with reductions ranging from 13.5 to 16.6%) and represented the most transplant procedures in the country. The most affected regions were the North and Northeast (with reductions of up to 47%) (Table 3).

Overall, we identified 21,339 transplant rejection episodes, divided into 16,576 in the pre-pandemic period and 4763 during the COVID-19 pandemic. The overall median age was 46 (34–58) years, with patients being predominantly male (60%) and white (68%). No significant differences existed between the pre-pandemic and COVID-19 patient characteristics (Table 4).

The proportion of rejections corrected for the number of transplants performed showed a median varying from 41 to 47% (Table 5).

A seasonal monthly plot of the data indicated a high rejection rate between 2018 and 2021 (Figure 3A).

We found a similar pattern in the normalized plot (Figure 3B). The Poisson model showed an 18% (IRR 1.18 [95% CI 1.01 to 1.37], *p* = 0.04) increase in the rate of rejection episodes during the COVID-19 pandemic (Table 6).

## 4. Discussion

The results show that the COVID-19 pandemic had a significant impact on transplants performed by SUS in Brazil. The total number of organ transplants performed in 2021 was similar to 2015, representing a regression to levels from six years ago. We also demonstrated a reduction in transplant activities across states, resulting in a higher concentration of transplant procedures in the Southeast by the end of 2021. We found a greater proportion of rejections occurring during the pandemic. Combined, these three findings (reduction in the number of transplants, concentration of transplant procedures, and higher rates of rejection episodes) could have a potential impact on the outcomes and expansion of transplant activities in Brazil.

Through SUS, Brazil has achieved nearly universal access to healthcare services across the country [8,9]. Considering Brazil’s population of about 211.8 million [10], SUS is one of the largest public healthcare systems in the world, and it is the primary care provider for 75% of the population [7]. The Brazilian organ transplant system was the largest public program for kidney transplants worldwide. More than 95% of the transplants in Brazil were performed by SUS, including reimbursement of hospitalization and immunosuppressive medication [11]. In this study, we analyzed the DATASUS database, which captures information about most procedures (transplants and complications) in Brazil. We expect a low rate of unreported procedures because of the link between the data and reimbursement. SUS healthcare providers must register details about the procedures and provide patient information using standardized forms [7].

The COVID-19 pandemic negatively impacted the number of solid organ transplants performed worldwide, with an overall reduction of 15.92% in 2020 (ranging from 1.24 to 66.7%) [12]. In Brazil, we found a reduction of 19.3% and 22.6% for the years 2020 and 2021, respectively. The most severely affected procedure was lung transplants, followed by kidney and pancreas (ranging from 2.3 to 51.3%). The least affected was liver transplants, with reductions of 7.7% and 14.3% for the years 2020 and 2021, respectively (Table 1). Another study performed in Brazil showed a similar finding of a more pronounced reduction in transplant activity for lungs and a limited reduction in liver transplants during the pandemic [4]. This could be explained by the unavailability of intensive care beds, fewer deceased brain trauma donors, and concerns about COVID-19 infection for solid organ transplant patients [4]. Transplant rates are returning to pre-pandemic levels in some countries, as a result of significant advances in the management of COVID-19, especially during the second year of the pandemic [13]. Despite these advances, Brazil faces difficulties returning to the pre-pandemic transplant levels.

The present study shows a concentration of transplant procedures in one region of the country. Before the pandemic, transplants were performed in the North and Northeast regions, but after the pandemic in 2021, transplants were concentrated in four States: São Paulo, Rio de Janeiro, Minas Gerais, and Parana. Similarly, another study showed a reduction in transplants, screening, and diagnostic tests in all Brazilian regions in 2020, with only the Southeast continuing to provide such care during the pandemic [7].

We hypothesized that there was an increase in the number of solid organ rejections during the COVID-19 pandemic. This increase may be related to changes in the immunosuppression regimen [14], combined with the prolonged interval between medical follow-up appointments imposed by the pandemic [15]. Previous case studies showed higher rejection rates during the pandemic, possibly related to missing clinic visits [16]. A study from Germany using nationwide administrative claim data showed an increase in kidney rejection rates from 14.4% to 22.7% between the years 2019 and 2020 [17]. Another study that performed kidney biopsies on those recovering from COVID-19 suggested a possible increased risk of rejection [18]. We found an increase in rejection rates, especially in the year 2021, which may reflect the effects of the pandemic in Brazil. These could be related to changes in immunosuppressive medications; the prolonged interval between medical appointments, as noted above; or to changes in the procurement of immunosuppressive medication by the government. In Brazil, immunosuppressive medications are provided by SUS. However, there have been reports of a shortage of transplant medication, which was particularly acute in 2017 and 2018. In the official publication of the government, a petition was published requesting information about the lack of tacrolimus and mycophenolate in 2017 and 2018. In this document, the government pledged to regularize the purchase of immunosuppressive medication by the end of 2018 [19].

We can speculate that extraordinary events such as the COVID-19 pandemic and the lack of immunosuppressive medication available in the public healthcare system could significantly impact rejection rates for solid organ transplant patients. The rate of rejection increased by 18% during the COVID-19 pandemic, reaching 47% (rejection corrected for number of transplants performed) in 2021. During 2018, there was also an increase in rejection rates, which could be related to the shortage of immunosuppressants available through SUS.

This study had some limitations. Although we cannot quantify the exact amount, we expect some underreporting in the DATASUS system. However, we expect this to be low because the system is linked to reimbursement. The total number of Brazilian transplants may be higher because we did not consider procedures undertaken by private healthcare providers. We cannot retrieve data on the immunosuppressive medications that patients were using, nor can we retrieve the histological type of the rejection (T-cell or antibody-mediated rejection). The frequency of rejection episodes was probably lower because the at-risk patients were not only recent transplant patients but also those in regular follow-up. Despite this limitation, correcting the rejection episodes by number of transplants performed enabled us to identify a value that could be compared over time.

In conclusion, data from a large, public organ transplant program showed that the COVID-19 pandemic had a significant impact on the transplants performed in Brazil’s Unified Healthcare System. The total number of organ transplants performed in 2021 represents a setback of six years. We demonstrated a higher concentration of transplants centered in the Southeast and a greater proportion of rejections during the pandemic. Combined, these three findings (reduction in the number of transplants, concentration of transplant activities, and more rejection episodes) could potentially impact the outcomes and expansion of transplant procedures in Brazil. We can speculate that extraordinary events such as the COVID-19 pandemic and a lack of immunosuppressants could significantly affect the rates of rejection in solid organ transplant patients. Additionally, data from the nationwide system can be used to monitor transplant activities throughout the year and by region. This data can also be used to monitor the results of transplant procedures or exceptional events that could affect transplant patient outcomes.

## Figures and Tables

**Figure 1 jcm-11-06581-f001:**
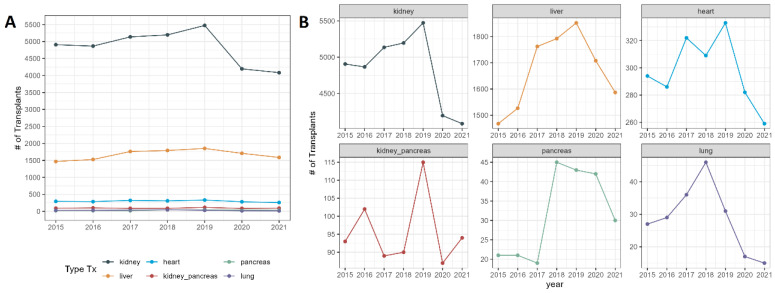
Total number of solid organ transplants performed in Brazil’s Unified Healthcare System (SUS) from 2015 to 2021. (**A**) All solid organ transplants; (**B**) individual graphs for each type of solid organ transplant. #: number.

**Figure 2 jcm-11-06581-f002:**
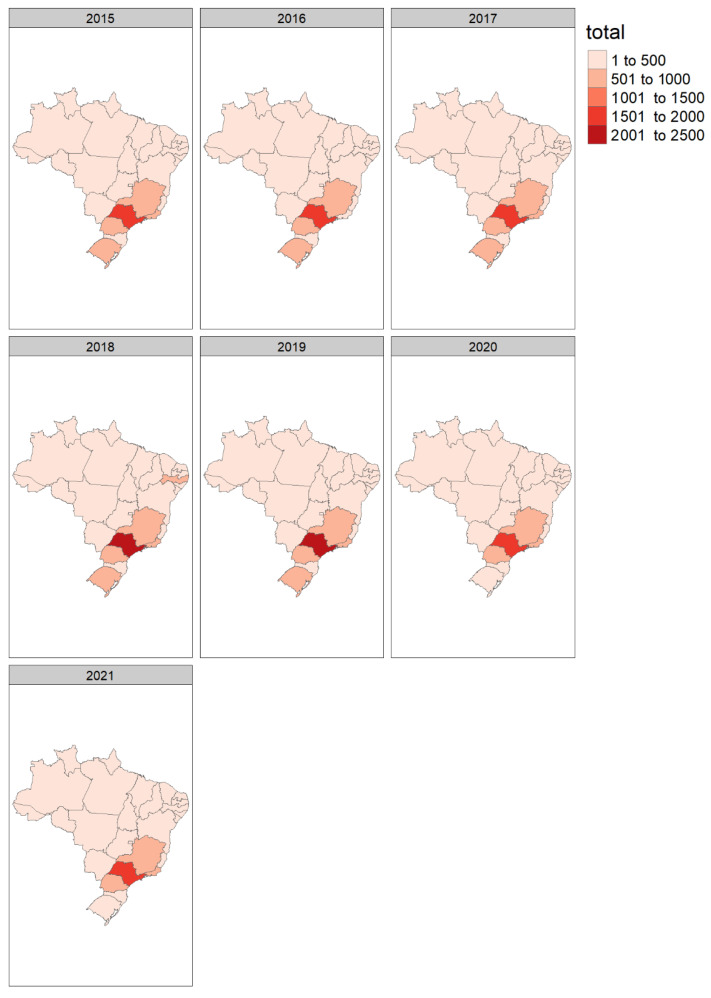
Total number of solid organ transplants performed in Brazil’s Unified Healthcare System (SUS) from 2015 to 2021 by state.

**Figure 3 jcm-11-06581-f003:**
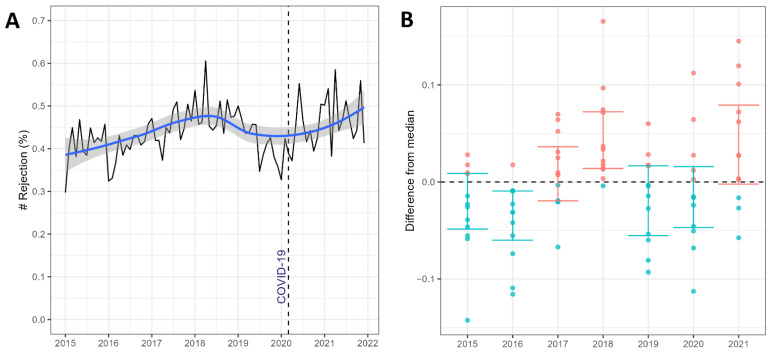
Total rejection episodes corrected for the number of transplants performed in Brazil’s Unified Healthcare System (SUS). (**A**) Monthly data plot divided into pre-pandemic (January 2015 to March 2020) and COVID-19 pandemic (March 2020 to December 2021) periods. (**B**) Normalized plot (difference from the median) per year from January 2015 to December 2021. #: number.

**Table 1 jcm-11-06581-t001:** Total number of solid organ transplants performed in Brazil’s Unified Healthcare System (SUS) from 2015 to 2021.

Characteristic	2015, *n* = 6809	2016, *n* = 6831	2017, *n* = 7364	2018, *n* = 7478	2019, *n* = 7847	2020, *n* = 6331	2021, *n* = 6069
Organ, *n* (%)							
Kidney	4906 (72)	4866 (71)	5136 (70)	5196 (69)	5473 (70)	4195 (66)	4084 (67)
Reduction *	-	-	-	-	-	**23.3%**	**25.4%**
Liver	1468 (22)	1527 (22)	1762 (24)	1792 (24)	1852 (24)	1708 (27)	1587 (26)
Reduction *	-	-	-	-	-	**7.7%**	**14.3%**
Heart	294 (4.3)	286 (4.2)	322 (4.4)	309 (4.1)	333 (4.2)	282 (4.5)	259 (4.3)
Reduction *	-	-	-	-	-	**15.3%**	**22.2%**
Kidney–pancreas	93 (1.4)	102 (1.5)	89 (1.2)	90 (1.2)	115 (1.5)	87 (1.4)	94 (1.5)
Reduction *	-	-	-	-	-	**24.3%**	**18.2%**
Pancreas	21 (0.3)	21 (0.3)	19 (0.3)	45 (0.6)	43 (0.5)	42 (0.7)	30 (0.5)
Reduction *	-	-	-	-	-	**2.3%**	**30.2%**
Lung	27 (0.4)	29 (0.4)	36 (0.5)	46 (0.6)	31 (0.4)	17 (0.3)	15 (0.2)
Reduction *	-	-	-	-	-	**45%**	**51.6%**

* Total transplant reduction compared to 2019.

**Table 2 jcm-11-06581-t002:** Number of solid organ transplants performed in Unified Health System (SUS) in Brazil divided into pre-pandemic (January 2015 to March 2020) and COVID-19 pandemic (March 2020 to December 2021) periods.

Characteristic	Overall *n* = 48,729	Pre-Pandemic *n* = 38,429	COVID-19 *n* = 10,300	*p*-Value
**Age (years), Median (IQR)**	49 (36–58)	49 (36–58)	49 (37–59)	0.066
**Sex, *n* (%)**				0.077
Female	18,280 (38)	14,339 (37)	3941 (38)	
Male	30,449 (62)	24,090 (63)	6359 (62)	
**Race, *n* (%)**				0.522
White	23,066 (57)	18,364 (58)	4702 (54)	
Black	4008 (9.9)	2913 (9.2)	1095 (12)	
Mixed	12,794 (32)	9897 (31)	2897 (33)	
Asian	591 (1.5)	518 (1.6)	73 (0.8)	
Indigenous	6 (<0.1)	4 (<0.1)	2 (<0.1)	
Not identified	8264	6733	1531	
**Tx Organ, *n* (%)**				<0.001
Kidney	33,856 (69)	26,987 (70)	6869 (67)	
Liver	11,696 (24)	8942 (23)	2754 (27)	
Heart	2085 (4.3)	1637 (4.3)	448 (4.3)	
Kidney–pancreas	670 (1.4)	512 (1.3)	158 (1.5)	
Pancreas	221 (0.5)	170 (0.4)	51 (0.5)	
Lung	201 (0.4)	181 (0.5)	20 (0.2)	
**Length of Stay (days), Median (IQR)**	10 (7–13)	10 (7–13)	9 (7–12)	<0.001
**Death, *n* (%)**	2121 (4.4)	1677 (4.4)	444 (4.3)	0.81

**Table 3 jcm-11-06581-t003:** Total number of solid organ transplants performed in Brazil’s Unified Healthcare System (SUS) from 2015 to 2021 by region.

Region	2015, *n* = 6809	2016, *n* = 6831	2017, *n* = 7364	2018, *n* = 7478	2019, *n* = 7847	2020, *n* = 6331	2021, *n* = 6069
Midwest	218	248	290	440	465	424	409
Reduction *	-	-	-	-	-	**8.8%**	**12%**
North	165	188	173	188	230	121	137
Reduction *	-	-	-	-	-	**47.3%**	**40.4%**
Northeast	1313	1232	1422	1492	1557	1032	1165
Reduction *	-	-	-	-	-	**33.7%**	**25.2%**
South	1629	1872	1899	1924	1798	1464	1192
Reduction *	-	-	-	-	-	**18.5%**	**33.7%**
Southeast	3418	3172	3474	3434	3797	3290	3166
Reduction *	-	-	-	-	-	**13.5%**	**16.6%**

* Total transplant reduction compared to 2019.

**Table 4 jcm-11-06581-t004:** Transplant rejection in Brazil’s Unified Healthcare System (SUS) divided into pre-pandemic (January 2015 to March 2020) and COVID-19 pandemic (March 2020 to December 2021) periods.

Characteristic	Overall, *n* = 21,339	Pre-Pandemic, *n* = 16,576	COVID-19, *n* = 4763	*p*-Value
**Age (years), Median (IQR)**	46 (31–58)	46 (32–58)	45 (30–57)	0.006
**Sex, *n* (%)**				
Female	8464 (40)	6570 (40)	1894 (40)	0.87
Male	12,875 (60)	10,006 (60)	2869 (60)	
**Race, *n* (%)**				
White	11,694 (68)	9126 (69)	2568 (64)	0.435
Black	1287 (7.5)	928 (7.1)	359 (8.9)	
Mixed	3998 (23)	2946 (22)	1052 (26)	
Asian	198 (1.2)	154 (1.2)	44 (1.1)	
Indigenous	2 (<0.1)	2 (<0.1)	0 (0)	
Not identified	4160	3420	740	
**Type Rejection, *n* (%)**				<0.001
Rejection kidney	13,424 (63)	10,452 (63)	2972 (62)	
Rejection liver	5128 (24)	3901 (24)	1227 (26)	
Rejection heart	2326 (11)	1835 (11)	491 (10)	
Rejection heart–lung	63 (0.3)	56 (0.3)	7 (0.1)	
Rejection pancreas	244 (1.1)	193 (1.2)	51 (1.1)	
Rejection lung	154 (0.7)	139 (0.8)	15 (0.3)	
**Length of stay (days), Median (IQR)**	6 (2–12)	6 (3–12)	5 (2–11)	<0.001
**Death, *n* (%)**	673 (3.2)	491 (3.0)	182 (3.8)	0.003

**Table 5 jcm-11-06581-t005:** Total rejection episodes corrected for number of transplants performed per year in Brazil’s Unified Healthcare System (SUS).

Rejection *	2015	2016	2017	2018	2019	2020	2021
Prop, Median (IQR)	0.42 (0.39–0.45)	0.41 (0.38–0.43)	0.45 (0.42–0.48)	0.47 (0.45–0.51)	0.43 (0.38–0.46)	0.42 (0.39–0.46)	0.47 (0.44–0.52)

* Proportion of rejections corrected for number of transplants performed.

**Table 6 jcm-11-06581-t006:** Poisson model of rejection episodes corrected for number of transplants performed by year in Brazil’s Unified Healthcare System (SUS).

Characteristic	IRR (95% CI)	*p*-Value
2015 year (Reference)		
2016	0.97 (0.88 to 1.06)	0.51
2017	1.10 (1.00 to 1.20)	0.051
2018	1.18 (1.08 to 1.29)	<0.001
2019	1.03 (0.94 to 1.12)	0.59
2020	0.92 (0.79 to 1.07)	0.30
2021	0.99 (0.83 to 1.18)	0.89
Reference pre-pandemic		
COVID-19 pandemic	1.18 (1.01 to 1.37)	0.040

## Data Availability

The databases used in this study are open source and made publicly available by the Brazilian government at https://datasus.saude.gov.br (accessed on 2 October 2022).

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
