# Peer review of "Impact of the COVID-19 Pandemic on Solid Organ Transplant and Rejection Episodes in Brazil’s Unified Healthcare System"

_jcm, 2022, doi:10.3390/jcm11216581_

Round 1

Reviewer 1 Report

Thank you for possibility to review this interesting paper

My comments 

  1. Please provide p values in the tables 
  2. Can you provide a separate analysis of rejections in newly transplanted patients in covid period?

Author Response

Thank you for possibility to review this interesting paper

My comments 

  1. Please provide p values in the tables 

We provided the p-value for the tables 2, 4 and 6. These tables compared baseline characteristics in the pre-pandemic versus COVID-19 period. We are not applying any statistical test in the tables 1 and 3, just presenting the total number of transplants, so the p-value does not apply. The same goes for the table 05, as we are presenting the total number of rejection episodes in this table.

  1. Can you provide a separate analysis of rejections in newly transplanted patients in covid period?

All the patients were newly transplanted patients, but the rejections episodes could occur in patients transplanted in late periods, which is a limitation of the study. Most of the rejection episodes occurred in newly transplanted patients, so we were unable to provide a separate analysis of rejection during COVID.

Reviewer 2 Report

Dear authors

greetings for your manuscript.

I have not major revision.

If hystological type of rejection is not avaible now I suggest to study it for a possible next manuscript . Because it could improve the quality of your alert  data on this import topic about rejection episodes in COVID pandemic period. 

1.The main question is if COVID pandemic period affected on rejection episodes for solid organ transplantation. 

2. The topic is original and relevant in the field of organ transplantation because monitoring the augment of rejection episode correlate with pandemic period. So it is a big alert for restarting follow up in the correct manner. 3. It is referred to a specific period of analysis of data. 

4. I suggested to add histological data for next manuscripts if it is possible.

5. In the discussion they assert that in the COVID period the lack of immunosuppressive medications impact the rates of rejection in solid organ transplantation.

6. I think all the references are appropriated.

7. In table 2 and 4 if they compare two grups I think they have to calculate statistical significance.

Thank you

Author Response

I have not major revision.

If hystological type of rejection is not avaible now I suggest to study it for a possible next manuscript . Because it could improve the quality of your alert  data on this import topic about rejection episodes in COVID pandemic period. 

Thank you for the suggestions. We will evaluate the histological type of rejections in a future manuscript.

1.The main question is if COVID pandemic period affected on rejection episodes for solid organ transplantation. 

Although the retrospective study design doesn’t prove the causation, an association could be demonstrated between COVID-19 pandemic and rejection. This raises the question for future studies.

  1. The topic is original and relevant in the field of organ transplantation because monitoring the augment of rejection episode correlate with pandemic period. So it is a big alert for restarting follow up in the correct manner.

Thank you for the comment

  1. It is referred to a specific period of analysis of data. 

We retrieved a 7-year database by downloading data from the DATASUS server. The pandemic period was defined as March 2020 to December 2021. The pre-pandemic period was defined from January 2015 to March 2020

  1. I suggested to add histological data for next manuscripts if it is possible.

Thank you for the suggestions. We will evaluate the histological type of rejections in a future manuscript.

  1. In the discussion they assert that in the COVID period the lack of immunosuppressive medications impact the rates of rejection in solid organ transplantation.

It was referred to as a pick of rejection episodes that occurs the year 2017-2018. We had a lack of immunosuppressors by the government this year. In the government's official journal, there was a petition requesting information about the lack of tacrolimus and mycophenolate in 2017 and 2018. Then, it could be related to a rise in rejection episodes. 

  1. I think all the references are appropriated.

Thank you.

  1. In table 2 and 4 if they compare two grups I think they have to calculate statistical significance.

We provided the p-values for the Table 02 and 04.

Reviewer 3 Report

COVID-19 had a negative impact on healthcare systems. The main questions raised by the Authors is whether COVID-19 influenced the number of transplantations of solid organs and on the number of rejection episodes of solid organ transplants in Brasil. In both studied fields, COVID-19 had a negative impact (i.e. number of transplantations decreased and the number of rejection episodes increased). These results could be expected; however the Authors provided evidence in these subjects based on Brasilian public transplant program, which is, nota bene, the largest public transplant program worldwide. Data on these issues from other countries have already been known, but this study enables to accumulate evidence by adding the data from the largest country of South America. As far as I know, to date, data from this country were unknown. The study is well designed. The methodology is correct. The text is well-written and clear. Conclusions are derivatives of presented data. The references are up-to-date and appropriate. Figures are correct. Table 2 requires correction, but other are correct. In my opinion, the article may be accepted after a minor revision. 

1. I suggest careful language edition, as individual mistakes can be noticed (e.g. in Line 15: "Brasil is the world's...," instead of "Brasil has the world's..."

2. In Tab. 2, races of patients are in Spanish, instead of English (pardo, amarela, indigena)

Author Response

Review 3

COVID-19 had a negative impact on healthcare systems. The main questions raised by the Authors is whether COVID-19 influenced the number of transplantations of solid organs and on the number of rejection episodes of solid organ transplants in Brasil. In both studied fields, COVID-19 had a negative impact (i.e. number of transplantations decreased and the number of rejection episodes increased). These results could be expected; however the Authors provided evidence in these subjects based on Brasilian public transplant program, which is, nota bene, the largest public transplant program worldwide. Data on these issues from other countries have already been known, but this study enables to accumulate evidence by adding the data from the largest country of South America. As far as I know, to date, data from this country were unknown. The study is well designed. The methodology is correct. The text is well-written and clear. Conclusions are derivatives of presented data. The references are up-to-date and appropriate. Figures are correct. Table 2 requires correction, but other are correct. In my opinion, the article may be accepted after a minor revision. 

  1. I suggest careful language edition, as individual mistakes can be noticed (e.g. in Line 15: "Brasil is the world's...," instead of "Brasil has the world's..."

Thank you for the comments. We provided an English revision of the manuscript.

  1. In Tab. 2, races of patients are in Spanish, instead of English (pardo, amarela, indigena)

We corrected the race according to the following reference:

Mays VM, Ponce NA, Washington DL, Cochran SD. Classification of race and ethnicity: implications for public health. Annu Rev Public Health. 2003;24:83-110. doi: 10.1146/annurev.publhealth.24.100901.140927.